

# A high-resolution synthetic bed elevation grid of the Antarctic continent

Felicity S. Graham[1], Jason L. Roberts[2,3], Ben K. Galton-Fenzi[2,3], Duncan Young[4], Donald Blankenship[4], and Martin J. Siegert[5]

[1]Institute for Marine and Antarctic Studies, University of Tasmania, Private Bag 129, Hobart, Tasmania 7001, Australia
[2]Department of the Environment, Australian Antarctic Division, Hobart, Tasmania, Australia
[3]Antarctic Climate and Ecosystems Cooperative Research Centre, Private Bag 80, Hobart, Tasmania 7001, Australia
[4]Institute for Geophysics, University of Texas at Austin, Austin, Texas 78758, USA
[5]Grantham Institute and Department of Earth Sciences and Engineering, Imperial College London, London SW7 2AZ, UK

*Correspondence to:* F. S. Graham (felicity.graham@utas.edu.au)

**Abstract.** Digital elevation models of Antarctic bed topography are heavily smoothed and interpolated onto low-resolution ($> 1$ km) grids as our current observed topography data are generally sparsely and unevenly sampled. This issue has potential implications for numerical simulations of ice-sheet dynamics, especially in regions prone to instability where detailed knowledge of the topography, including fine-scale roughness, is required. Here, we present a high-resolution ($100$ m) synthetic bed elevation terrain for the whole Antarctic continent. The synthetic bed surface preserves topographic roughness characteristics of airborne and ground-based ice-penetrating radar data from the Bedmap1 compilation and the ICECAP consortium. Broad-scale features of the Antarctic landscape are incorporated using a low-pass filter of the Bedmap2 bed-elevation data. Although not intended as a substitute for Bedmap2, the simulated bed elevation terrain has applicability in high-resolution ice-sheet modelling studies, including investigations of the interaction between topography, ice-sheet dynamics, and hydrology, where processes are highly sensitive to bed elevations. The data are available for download at the Australian Antarctic Data Centre (doi:10.4225/15/57464ADE22F50).

## 1 Introduction

Estimates of mass loss from the Antarctic and Greenland Ice Sheets are associated with the largest degree of uncertainty in projections of sea level rise over the coming century (Church et al., 2013). As the most vulnerable regions of the Antarctic Ice Sheet are grounded below sea level, the ice-sheet response to climate warming will be determined by dynamics operating at the grounding line (Schoof, 2007b; Drouet et al., 2013). For regions grounded below sea level and where the bed topography slopes downward into the interior of the ice-sheet, Marine Ice Sheet Instability (MISI) could occur, leading to increased ice flow, thinning, and rapid glacier retreat (Weertman, 1974; Thomas et al., 2004; Schoof, 2007a; Durand et al., 2009; Goldberg et al., 2009; Favier et al., 2014; Joughin et al., 2014). It follows that bed elevation is one of the most important controls in modelling ice-sheet dynamics and constraining estimates of future sea level rise.



Concerted international efforts over recent decades have vastly increased the scope and density of bed elevation measurements in Antarctica (Lythe et al., 2001; Le Brocq et al., 2010; Fretwell et al., 2013). These data have been used to improve the fidelity of gridded digital elevation models (DEMs) spanning the whole Antarctic continent. Building on a 5 km gridded bed elevation DEM (Lythe et al., 2001; Le Brocq et al., 2010), the most recently compiled Antarctic bed topography dataset,

Bedmap2, is available at 1 km resolution, having been generated from over 25 million measurements (Fretwell et al., 2013). Nevertheless, much of the Antarctic continent is difficult to access logistically and remains poorly sampled. In such regions, bed elevation DEMs rely on heavy smoothing and interpolation onto low-resolution grids, resulting in geometric inconsistencies that carry through numerical simulations of ice dynamics (Warner and Budd, 2000; Fürst et al., 2015; Gasson et al., 2015). Uncertainties in bed elevation are particularly problematic given that for much of the Antarctic Ice Sheet the simulated

large-scale velocity field depends only on the local scale details of the geometry and boundary conditions due to the elliptic nature of the governing equations for ice flow.

Recent effort has focussed on understanding the impact of low-resolution bed elevation data on ice mass-flux. Durand et al. (2011) performed a sensitivity analysis of an outlet glacier susceptible to MISI, demonstrating that a minimum of 1 km spatial resolution in bed topography is required for accurate estimates of ice mass flux. However, bed elevation data of a higher

resolution than 1 km may be necessary in some applications to capture both the channelised landscape that guides glacier flow and the fine-scale roughness that impacts basal sliding (Goff et al., 2014). Importantly, the question remains as to the minimum degree of spatial resolution required in bed topography DEMs to accurately model ice flow.

The purpose of this study is to generate a high-resolution synthetic bed topography dataset for Antarctica (HRES) with small-scale roughness incorporated, matched to that measured in available radar transects. The generation of HRES relies on

bed elevation data from the ICECAP airborne radar survey and the Bedmap1 compilation where they are available at high-resolution. HRES covers the same domain as Bedmap2 and is available at a spatial resolution of 100 m. HRES will be valuable for sensitivity studies investigating the impact of bed elevation resolution and roughness on ice-sheet dynamics, including the interaction with subglacial hydrology (Goff et al., 2014), and especially in the vicinity of grounding lines.

## 2   Data synthesis

A two-step approach was used to generate the high-resolution synthetic bed elevation terrain, HRES. We first simulated a non-conditional "roughness" terrain (i.e., a stochastic realisation of "roughness" that does not necessarily honour the exact values of the original data) using high spatial resolution radar data obtained from the 2009-2012 ICECAP campaigns (Roberts et al., 2011; Young et al., 2011; Wright et al., 2012) and the Bedmap1 compilation (Lythe et al., 2001). The locations of the data included in this step are illustrated in Fig. 1. The ICECAP bed elevation data are measured using a High-Capability Radar

Sounder (HiCARS) high bandwidth airborne ice penetrating radar (Peters et al., 2005); the Bedmap1 compilation combines data from multiple airborne and ground based radar sounding campaigns, from a variety of systems. Our method for the generation of the roughness terrain can easily incorporate additional bed elevation data, as they become available. Once generated, the roughness terrain was high-pass filtered using a gaussian kernel with a 5 km 1/2 power cutoff.





Second, the Bedmap2 bed topography DEM was low-pass filtered, using a low-pass gaussian kernel with a 5 km 1/2 power cutoff. The two filtered terrains were combined (preserving all wavelengths of the original datasets), resulting in the high-resolution bed topography, HRES. In the following sections, we provide a detailed outline of the methods used to generate the roughness terrain and to compile the final synthetic bed topography dataset. The 'pseudo' algorithm for the generation of HRES is provided in Appendix A.

## 2.1 Roughness terrain synthesis

Ideally, the spatial covariance characteristics of the non-conditional roughness terrain (the high frequency component of the synthetic topography dataset) should match those of the ICECAP and Bedmap1 datasets. The method of Cholesky decomposition of the observed covariance matrix can be used to produce such correlated data (Davis, 1987). Specifically, the positive definite covariance matrix $\mathbf{C}$ calculated from the ICECAP and Bedmap1 datasets can be decomposed into lower $\mathbf{L}$ and upper $\mathbf{U}$ triangular matrices

$$\mathbf{C} = \mathbf{L}\mathbf{U}, \tag{1}$$

where $\mathbf{L}$ has real and positive diagonal entries and $\mathbf{U}$ is the conjugate transpose of $\mathbf{L}$. This method results in a unique decomposition for positive definite matrix $\mathbf{C}$.

Now, given a vector $\mathbf{z}$ of uniformly distributed random numbers with zero mean, we find

$$\mathrm{Cov}(\mathbf{L}\mathbf{z}) = E[(\mathbf{L}\mathbf{z})(\mathbf{L}\mathbf{z})^T] = E(\mathbf{L}\mathbf{z}\mathbf{z}^T\mathbf{U}) = \mathbf{L}\mathbf{I}\mathbf{U} = \mathbf{C}. \tag{2}$$

So, the product $\mathbf{L}\mathbf{z}$ can be used to construct a non-conditional realisation of bed topography, the covariance structure of which is consistent with that of the ICECAP and Bedmap1 datasets. We next describe how the covariance matrix $\mathbf{C}$ and resulting simulated roughness terrain are calculated.

### 2.1.1 Covariance structure

In order to perform the Cholesky decomposition from Eq. (1), we first calculated the covariance distribution for the ICECAP and Bedmap1 datasets. The along-track covariance distribution for each flight or traverse line was estimated using 16 km sliding windows with 8 km offsets. For each window, the along-track data were averaged into 100 m bins and the following exponential decay model was fitted (Goff and Jordan, 1989):

$$C = A \exp\left(-\frac{d}{D}\right), \tag{3}$$

where $C$ is the covariance, $d$ the along track distance, $A$ is the topographic variance, and $1/D$ is the e-folding distance. Both $A$ and $D$ are free parameters obtained by linear least squares data fitting. To ensure that the data density was approximately consistent for each calculation of the covariance distribution, windows were included only if data were present in at least 90% of the 100 m bins. Additionally, data were omitted from the calculation if $A < 0$, $A > 500$, or the ratio of $A$ to the maximum



covariance in any 100 m bin was outside the range $[0.33, 3]$ (the latter condition ensured a reasonable fit to the exponential model).

A total of 9272 points satisfied the criteria for inclusion in the calculation of the non-conditional roughness terrain (Fig. 1).

### 2.1.2 Cholesky decomposition

The covariance matrix $\mathbf{C}$ defined by the exponential model in Eq. (3) is necessarily symmetric and positive definite. As such, Cholesky decomposition can be applied to $\mathbf{C}$ as per Eq. (1) to obtain the lower triangular matrix $\mathbf{L}$ and its conjugate transpose $\mathbf{U}$.

For a box with 8 km side lengths and easting and northing coordinates centred on each of the 9272 valid data points from Sect. 2.1.1, a covariance matrix $\mathbf{C}$ was calculated using coefficients $A$ and $D$ from Eq. (3), and with $d$ varying appropriately.

Cholesky decomposition was applied to each of these covariance matrices $\mathbf{C}$, yielding matrices $\mathbf{L}$.

Next, a uniform random matrix with zero mean was calculated over a spatial domain representing the whole of Antarctica – the same spatial domain as Bedmap2 – with 100 m resolution. For each point in this grid that did not have a corresponding covariance data point (the majority), the local Cholesky decomposition matrix $\mathbf{L}$ was generated as the inverse distance squared weighted average of the 20 closest Cholesky decompositions. The choice of 20 inverse distance squared weighted points

minimised artefacting associated with sparse data.

Finally, the Cholesky decomposition matrix $\mathbf{L}$ was multiplied by the uniform random matrix, resulting in a synthetic Cholesky decomposition roughness terrain (CDRT) of random data with spatial covariance structure consistent with that of the original ICECAP and Bedmap1 data. Note that although CDRT is one realisation of an infinite number of unique roughness terrains, this realisation suffices for our purposes (namely, to generate a synthetic bed topography dataset suitable for

investigating the impact of resolution on ice-sheet dynamics). The calculation of CDRT was spatially independent for each data point, so computational parallelism through the use of OpenMP directives was utilised to reduce computational time (which was on the order of 2000 CPU hours).

### 2.2 Compilation of HRES

Due to the statistical properties of large samples of distributions, the bed elevation extrema from CDRT were -72 897 and

70 838 m, well outside the observed range from the ICECAP/Bedmap1 measurements of -3373 – 3380 m. To address this, we defined a scaling factor based on a comparison of roughness values from the original and simulated datasets. Roughness $G$ is defined as the root mean squared deviation between points of detrended bed elevation $z$ separated by a lag ($\triangle x$; Shepard et al., 2001),

$$G = \sqrt{\frac{1}{n} \sum_{i=1}^{n} [z(x_i) - z(x_i + \triangle x)]^2}. \qquad (4)$$

We used a lag of 1600 m, consistent with that used by Gooch et al. (2016). Roughness values were calculated using Eq. (4) for each of the ICECAP/Bedmap1 flight lines separately, and for the points in CDRT that overlaid these data. A linear fit to



the spread of roughness values from ICECAP/Bedmap1 and CDRT yielded a slope of 14.42 and $R^2$ value of 0.49 (significant at the 95% confidence interval using a two-sided Student's t-test) for the correlation between the observed and predicted values (Fig. 2a). We used the median value of the ratio between ICECAP/Bedmap1 and CDRT roughnesses – approximately 22.87 – to uniformly scale CDRT. This median value was close to the ratio of CDRT to ICECAP/Bedmap1 extrema of 21.29.

Once scaled, CDRT was high-pass filtered using a gaussian kernel with a 5 km 1/2 power cutoff. The corresponding low-pass gaussian kernel was used to filter the Bedmap2 DEM, which was first interpolated to the same 100 m grid as CDRT. The two filtered datasets were then added to produce HRES.

## 3   Results

The HRES terrain is plotted alongside Bedmap2 in Fig. 3. HRES bed elevations range from -8848 to 4008 m: within 25% and

10% of the corresponding bounds in Bedmap2, which are -7054 and 3972 m, respectively. The very low bed elevations in both HRES and Bedmap2 lie offshore. The low frequency components of the two datasets are identical, so the difference between them ($D = \text{Bedmap2} - \text{HRES}$) is essentially a measure of the CDRT roughness introduced in HRES.

HRES was generated from a non-conditional simulation of the ICECAP/Bedmap1 data that is unlikely to honour the exact values of the underlying data. For this reason, the magnitude of the differences between HRES and Bedmap2 is not necessarily

the most robust measure of the quality of HRES. Instead, the extent to which the distribution of $D$ differs from a normal distribution provides an indication of the fidelity of HRES to the original ICECAP/Bedmap1 data. We calculate the deviation of the distribution of $D$ from the normal distribution using the D'Agostino-Pearson $K^2$ test (D'Agostino et al., 1990). The test statistic $K^2$ has is approximately chi-squared distributed with two degrees of freedom. $K^2$ calculates deviation from normality as a result of skewness and/or kurtosis, and is defined as

$$K^2 = Z^2(\sqrt{b_1}) + Z^2(b_2), \tag{5}$$

where $Z(\sqrt{b_1})$ is a test of skewness ($\sqrt{b_1}$), and $Z(b_2)$ is a test of kurtosis ($b_2$). The test statistic is calculated for each of the Antarctic drainage basins defined using ICESat altimetry (Table 1; Zwally et al., 2012), and the corresponding distributions of $D$ are compared with the normal distribution in Fig. 4.

The distribution of $D$ deviates most from the normal distribution in regions where more ICECAP/Bedmap1 data are avail-

able and/or meet the criteria for inclusion in the simulation of CDRT. In East Antarctica, these are ICESat basins 12-17, which include areas of Wilkes Land and the northern tail of the Transantarctic Mountains, and an area within Palmer Land in the Antarctic Peninsula (basin 24). Basin 21 is the only basin that is not statistically significantly different from the normal distribution at the 95% confidence interval. Nevertheless, the distribution of $D$ is generally closer to the normal distribution in regions with the poorest ICECAP/Bedmap1 data coverage, including much of West Antarctica (basins 1, 20-23, and 25, which

encompass Marie Byrd Land and the Siple Coast, Ellsworth Land, and the Filchner and Ronne Ice Shelf) and basins 5-9 in Queen Maud Land, East Antarctica. Spikes in the distribution of $D$ in basins 18 and 19 delineate smooth terrain over the Ross Ice Shelf from rougher, continental terrain.



Differences between ICECAP/Bedmap1 data points and the corresponding overlay points in HRES and Bedmap2 along 18 selected ICECAP/Bedmap1 flight or traverse lines are compared in Fig. 5. These flight/traverse lines encompass a range of landscapes, from smooth subglacial basins to high-elevation highlands. For over half of the selected flight/traverse lines, the along-track roughness values from HRES are within 20% of the corresponding roughness values from ICECAP/Bedmap1.

Flight lines O, P, and R show the poorest agreement in roughness between HRES and ICECAP/Bedmap1, with roughness values deviating by more than 50% of the higher value in each case. However, flight/traverse lines O, P, and R are derived from regions with a paucity of high-resolution data available for inclusion in the generation of CDRT (flight lines O, P, and R were themselves not included in the generation of CDRT for this reason). As expected, where Bedmap2 data are in better agreement with ICECAP/Bedmap1, the normalised along-track root mean square error between HRES and ICECAP/Bedmap1

is minimised (Table 2). This relationship holds independent of the underlying terrain roughness.

## 4  Errors and uncertainties

Sources of uncertainty exist in the datasets, methods, and processes used to generate HRES. We do not quantify these errors explicitly because HRES has been generated predominantly for investigating the impact of resolution and roughness on ice-sheet dynamics, rather than as a realistic, specific representation of Antarctic bed topography. Nevertheless, the following

inconsistencies in the generation of HRES should be taken into account:

(i) the roughness terrain incorporated in HRES is a non-conditional simulation of high-resolution flight/traverse line data from the ICECAP and Bedmap1 compilations, that are themselves sparsely available over the Antarctic continent (Fig. 1). The flight/traverse line data have associated errors from instrumentation and processing (e.g., Peters et al., 2005) – these errors will propagate through the simulation of HRES;

(ii) the Bedmap2 DEM, of which the low-pass component is included in the generation of HRES, suffers artefacting through the particular gridding and interpolating methods used compared with other ice thickness interpolation methods, especially in regions with no nearby measurements (Roberts et al., 2011);

(iii) the non-conditional simulation technique based on the Cholesky decomposition of ICECAP/Bedmap1 covariances makes a number of assumptions that influence the outcome bed elevations (notably, that the original data are isotropic and that

high frequency noise is normally distributed); and

(iv) HRES is simulated using data – ICECAP, Bedmap1, and Bedmap2 – that are not independent.

We refer to the original datasets and methods papers for a more detailed discussion of errors inherited by the HRES dataset from the underlying terrains (Alabert, 1987; Davis, 1987; Bourgault, 1997; Lythe et al., 2001; Le Brocq et al., 2010; Young et al., 2011; Fretwell et al., 2013).





## 5 Data Provenance and Structure

The result of this study is a 100 m resolution gridded Antarctic bed elevation terrain – namely, HRES – that has been made available for download at the Australian Antarctic Data Centre (data.aad.gov.au; doi:10.4225/15/57464ADE22F50). HRES combines a high frequency non-conditional simulation of bed elevation with the low frequency component of the Bedmap2

bed elevation terrain. This dataset is available in NetCDF classic format on a 100 m resolution grid in a Polar Stereographic Projection (Central Meridian $0°$, Standard Parallel $71°$S) with respect to the WGS84 geoid. The 100 m grid is 66661 rows by 66661 columns, where the corner of the lower left cell is located at a polar stereographic easting and northing of $-3333000$ m and $-3333000$ m, respectively. The value for missing data is -9999. The file size is approximately 17 GB.

## 6 Conclusions

We simulated a high-resolution (100 m) bed topography dataset of the entire Antarctic continent – HRES. HRES combines a high frequency "roughness" terrain based on the covariance properties of ice penetrating radar derived bed elevation data with low frequency data from the Bedmap2 bed elevation dataset. The final HRES terrain is made available in netCDF format.

An alternative method for the simulation of high (250m) resolution bed elevation data has recently been applied to the Thwaites Glacier region (Goff et al., 2014). This method combines both conditional and non-conditional simulations of a

15 range of data with the intent to avoid the inconsistencies and artefacting introduced through interpolation techniques such as kriging. The resulting terrain is of sufficient resolution to allow characterisation of the subglacial landforms and landscape of the Thwaites Glacier, which will lead to improved understanding of ice flow and its sensitivities to external forcing. However, the methods used to produce this terrain rely on a higher data density than is available for most of Antarctica.

HRES is not intended as a realistic depiction of high-resolution Antarctic bed topography and is, therefore, not meant as a

20 substitute for datasets such as Bedmap2. Despite this, HRES has applicability in high-resolution numerical modelling studies of Antarctic ice-sheets. For example, this dataset will allow the investigation of bed topography resolution needed for stable ice flow in numerical ice-sheet models. The dataset will also emphasise regions where high-resolution bed elevation data are needed, in order to target efforts in data collection. The Cholesky decomposition method used to simulate HRES may be extended to isotropic fields in other areas of research where observations are sparse, such as in the mapping of bathymetry in

oceanographic studies, or of roughness in the topography under ice shelves.

## Appendix A: Pseudo code for the non-conditional simulation

```
FOR all ICECAP/Bedmap1 flight lines with bed elevations
  Calculate 16km sliding window composed of 100m bins
  IF data present in at least 90% of bins THEN
    Fit exponential decay model, calculating
    coefficients A, D, for along track distance d
    IF (A<0 or A>500 or
        A/max[covariance] not in [0.33, 3.0]) THEN
```



```
       Discard window
     END IF
    END IF
    Move 8km to calculate next 16km window
END FOR
   FOR all 9272 valid points with coefficients A, D, and d
     Calculate covariance matrix C within 8km x 8km
         box and apply Cholesky decomposition,
         obtaining lower triangular matrix L
END FOR
   FOR all grid points on a 100m resolution
      mesh covering spatial domain of Antarctica
     Calculate inverse distance squared weighted
     Cholesky decomposition matrix L from existing
L matrices
     Matrix multiply L by random uniform matrix z,
     obtaining CDRT
   END FOR
   Add CDRT and low-pass filtered Bedmap2
bed elevation terrain
```

*Acknowledgements.* The authors thanks Richard Coleman and David E. Gwyther for constructive feedback. This research is supported under the Australian Research Council's Special Research Initiative for Antarctic Gateway Partnership SR140300001. The project is part of an ongoing ICECAP collaboration between Australia, the USA, and the UK, and is supported by the Australian Antarctic Division projects 3013, 4077 and 4346, the Antarctic Climate and Ecosystems Cooperative Research Centre, NSF grants PLR-0733025 and PLR-1143843, and CDI-0941678, NASA grants NNG10HPO6C and NNX11AD33G (Operation Ice Bridge and the American Recovery and Reinvestment Act), NERC grant NE/D003733/1, the G. Unger Vetlesen Foundation, the Jackson School of Geosciences, University of Texas, and the British Council Global Innovation Initiative Award.



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







**Figure 1.** Locations of ICECAP/Bedmap1 bed elevation data included in the synthesis of CDRT. Data are coloured by the natural log of the amplitude coefficient in the covariance data fit, namely $\log A$ in Eq. (3).





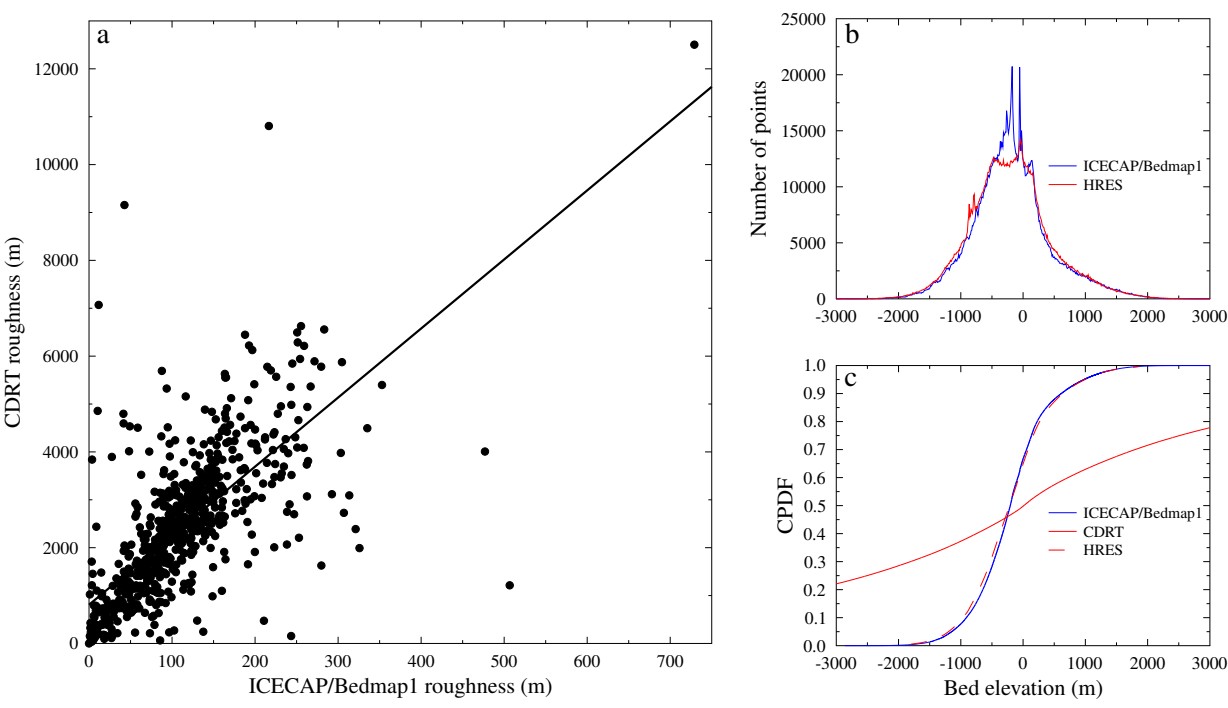

**Figure 2. (a)** Roughness values for each of the ICECAP/Bedmap1 compilations ($x$-axis) and the corresponding overlay points in Cholesky decomposition roughness terrain (CDRT; $y$-axis) calculated from Eq. (4). The fitted line is calculated using linear least squares. **(b)** Binned distribution of bed elevation points from the ICECAP/Bedmap1 compilations and the corresponding overlay points in HRES. **(c)** Cumulative probability density function of bed elevations.





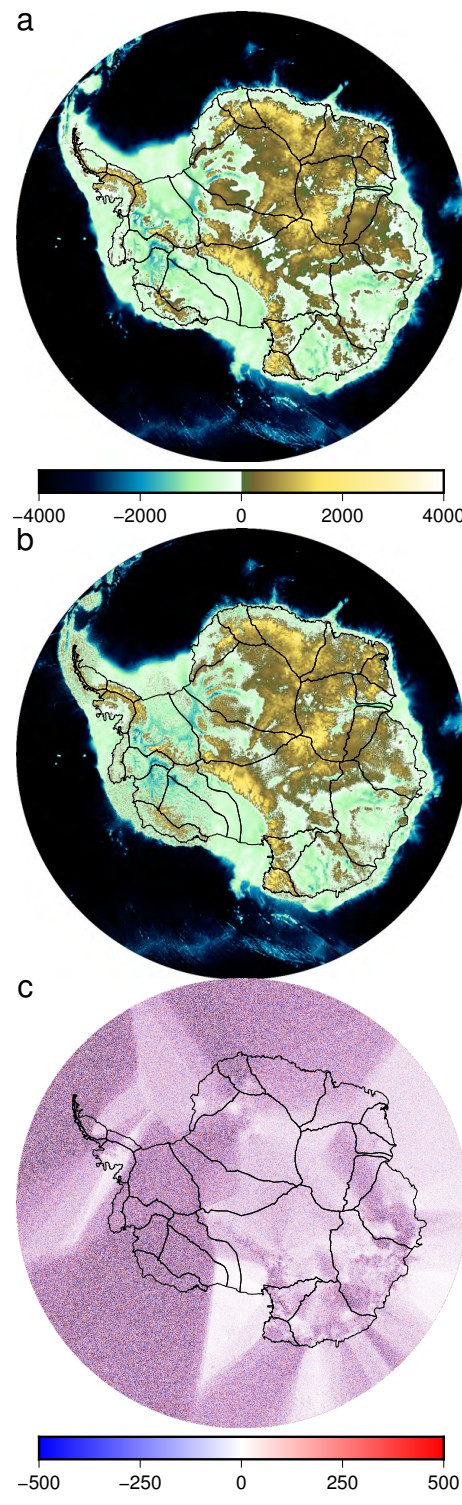

**Figure 3. (a)** Bedmap2 bed elevation (m), **(b)** HRES bed elevation (m), and **(c)** difference between Bedmap2 and HRES bed elevations (m).

The drainage basins in each panel (black lines) are taken from the Goddard Ice Altimetry Group from ICESat data (http://icesat4.gsfc.nasa.

gov/cryo_data/ant_grn_drainage_systems.php)           **14**





**Figure 4.** Distribution of HRES over the Antarctic drainage divides from the Goddard Ice Altimetry Group from ICESat data. Basins 2-17 are in East Antarctica, basins 1, and 18-23 are in West Antarctica, and the remaining basins are located in the Antarctic Peninsula (see Fig. 3). In each panel, the blue binned data are from HRES and the red dashed line shows the normal distribution from the given mean and standard distribution of the HRES data.

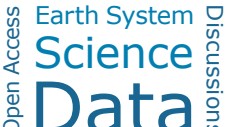

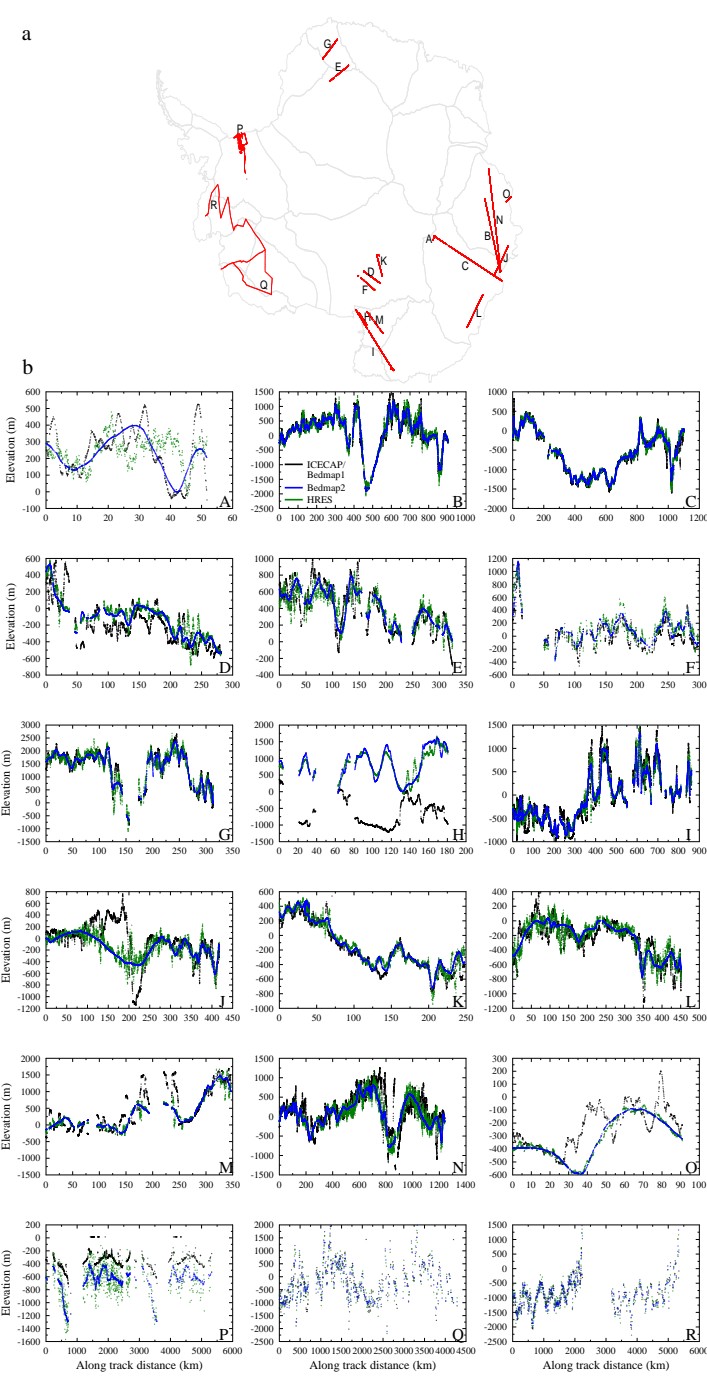

**Figure 5. (a)** Locations of selected flight lines from the ICECAP/Bedmap1 compilations. **(b)** Along track bed elevations from ICE-CAP/Bedmap1 (black) and corresponding overlay points from Bedmap2 (blue) and HRES (green). The x-axis shows the along track distance (km) from the first point in the flight line.


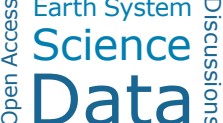

**Table 1.** Statistics from the D'Agnostino-Pearson $K^2$ normality test Eq. (5) for each of the drainage basins 1-27 in Fig. 4. The $\sqrt{b_1}$ and $b_2$ statistics are the bases for tests of skewness and kurtosis, respectively. For a normal distribution, $K^2$ is approximately chi-distributed with two degrees of freedom.

| Basin | $\sqrt{b_1}$ | $b_2$ | $Z(\sqrt{b_1})$ | $Z(b_2)$ | $K^2$ | $p$ |
|---|---|---|---|---|---|---|
| 1 | -0.01 | 3.09 | -2.16 | 14.82 | 224.33 | 0.00 |
| 2 | 0.01 | 4.84 | 2.95 | 193.11 | 37300.76 | 0.00 |
| 3 | 0.00 | 6.34 | 0.18 | 347.69 | 120888.51 | 0.00 |
| 4 | 0.00 | 4.81 | 0.57 | 115.23 | 13279.22 | 0.00 |
| 5 | -0.02 | 4.37 | -4.95 | 83.48 | 6992.95 | 0.00 |
| 6 | -0.02 | 3.58 | -7.44 | 77.31 | 6031.82 | 0.00 |
| 7 | -0.02 | 3.93 | -5.45 | 93.50 | 8771.66 | 0.00 |
| 8 | 0.06 | 3.85 | 10.61 | 49.83 | 2595.36 | 0.00 |
| 9 | 0.03 | 4.03 | 5.20 | 57.94 | 3384.57 | 0.00 |
| 10 | 0.03 | 4.70 | 12.62 | 187.51 | 35319.39 | 0.00 |
| 11 | 0.01 | 3.91 | 1.47 | 67.42 | 4547.69 | 0.00 |
| 12 | 0.01 | 6.68 | 2.74 | 257.46 | 66290.96 | 0.00 |
| 13 | 0.02 | 8.56 | 8.39 | 365.49 | 133654.10 | 0.00 |
| 14 | 0.02 | 5.51 | 7.82 | 207.25 | 43014.01 | 0.00 |
| 15 | 0.12 | 3.87 | 18.39 | 46.05 | 2458.58 | 0.00 |
| 16 | 0.02 | 7.31 | 4.26 | 160.71 | 25845.09 | 0.00 |
| 17 | 0.09 | 6.20 | 52.45 | 387.82 | 153152.39 | 0.00 |
| 18 | 0.02 | 5.14 | 4.87 | 140.09 | 19650.24 | 0.00 |
| 19 | -0.01 | 3.62 | -3.41 | 66.41 | 4422.35 | 0.00 |
| 20 | 0.00 | 3.20 | 0.58 | 18.21 | 331.76 | 0.00 |
| 21 | 0.01 | 3.02 | 1.13 | 2.18 | 6.03 | 0.05 |
| 22 | -0.00 | 3.14 | -0.63 | 12.13 | 147.46 | 0.00 |
| 23 | 0.01 | 3.05 | 1.71 | 3.45 | 14.82 | 0.00 |
| 24 | -0.06 | 7.04 | -10.77 | 139.75 | 19646.67 | 0.00 |
| 25 | 0.00 | 3.34 | 0.32 | 11.44 | 130.89 | 0.00 |
| 26 | -0.01 | 4.88 | -1.75 | 67.36 | 4541.02 | 0.00 |
| 27 | -0.02 | 4.17 | -2.00 | 40.79 | 1667.76 | 0.00 |



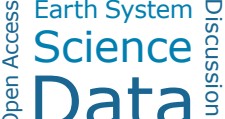

**Table 2.** Along track roughness (m) from the ICECAP/Bedmap1 flight lines in Fig. 5 and the corresponding roughness values from the HRES and Bedmap2 overlay points. Root mean square errors (rmse; m) between the ICECAP/Bedmap1 data and the corresponding HRES and Bedmap2 data were normalised by the square root of the number of points in each track.

| flight line | ICECAP/Bedmap1 | HRES | | Bedmap2 | |
|---|---|---|---|---|---|
| | roughness (m) | roughness (m) | rmse (m) | roughness (m) | rmse (m) |
| A | 134.8 | 82.7 | 6.6 | 41.5 | 3.8 |
| B | 166.2 | 177.8 | 2.4 | 64.9 | 1.5 |
| C | 83.6 | 77.8 | 1.0 | 30.9 | 0.8 |
| D | 107.3 | 85.9 | 3.2 | 25.3 | 3.1 |
| E | 121.9 | 113.5 | 3.4 | 36.2 | 2.7 |
| F | 134.5 | 93.1 | 5.5 | 74.6 | 4.0 |
| G | 250.9 | 267.9 | 5.5 | 150.2 | 3.2 |
| H | 143.7 | 162.2 | 41.0 | 154.6 | 42.4 |
| I | 159.3 | 132.1 | 2.5 | 79.6 | 2.1 |
| J | 118.9 | 117.2 | 5.2 | 25.4 | 5.2 |
| K | 72.7 | 75.4 | 1.8 | 32.1 | 1.4 |
| L | 111.1 | 104.4 | 2.5 | 21.1 | 2.2 |
| M | 208.8 | 125.7 | 8.6 | 39.0 | 8.3 |
| N | 158.7 | 125.6 | 3.5 | 274.0 | 3.4 |
| O | 75.7 | 30.2 | 5.9 | 21.5 | 6.1 |
| P | 38.6 | 169.8 | 12.7 | 4.8 | 11.7 |
| Q | 104.6 | 150.9 | 20.2 | 10.3 | 17.2 |
| R | 10.0 | 270.0 | 11.8 | 34.7 | 9.5 |