# Peer review of "A high-resolution synthetic bed elevation grid of the Antarctic continent"

_Earth System Science Data, 2016_

## Referee Comment (RC1) · Anonymous Referee #1 · 23 Aug 2016

Review of Graham et al

**General comments**

The authors have produced a synthetic bed topography for Antarctica aimed at assessing the "interaction between topography, ice-sheet dynamics and hydrology". This is a worthy objective because, despite over half a century of airborne campaigns, there are still large swathes of Antarctica absent of any data on bed properties. The data set could be of value for testing the sensitivity of numerical models to bed topography but I have two general concerns about its use for this. First, it is evident that "topographic variance" is not a smoothly varying function spatially. This is problematic because the high-frequency component of the topography has only been sampled for a small fraction of the continent (Fig 1). The high frequency characteristics of almost all of West Antarctica and most of East, remain uncharted. In addition, the fraction sampled does not incorporate many of the marine sectors (particularly in West Antarctica) that are highlighted in the introduction as motivation for the work.

Second, the authors refer to "topographic roughness" without being clear about precisely what length scale this relates to. Inherently this length scale is determined by the along track sampling properties of the radar system rather than any underlying geophysical criteria. This is problematic because a key uncertainty in future projections is basal traction (Ritz et al, 2015), which is modulated by metre scale roughness. It would appear that HRES provides no information on roughness at this scale or at any scale below 200 m (given 100 m bin size). While this is shorter wavelength than BEDMAP2 it remains to demonstrated that it is adequate to elucidate the role and/or importance of "bed roughness" on ice dynamics.

These issues are challenging to address, requiring greater data coverage, which does exist, and sensitivity studies with an ice sheet model to scales of basal topographic variability. It is beyond the scope of this study to do this but the authors, nonetheless, need to include consideration of these points in the paper.

**Specific comments**

p2, l7 "heavy smoothing" is non scientific terminology. What is heavy? It doesn't have a mass.

p2, l8 poor phrasing.

p2 l16-17 ditto

p4, l15 artefacting not a word

Fig 3. Figs a and b essentially identical (to the eye) and provide no real insight: they are identical to the BEDMAP2 topography at the scale plotted. Fig c appears to have had something horrible go wrong with the colour table/conversion. I have no idea what value the purple colours are. This figure needs redrafting to provide some useful info.

Fig 5. Even when zoomed to larger than the printed page I struggled to read the numbers and see the detail in the 18 graphs in this figure.

**Reference**

Ritz, C., Edwards, T. L., Durand, G., Payne, A. J., Peyaud, V., and Hindmarsh, R. C. A.: Potential sea-level rise from Antarctic ice-sheet instability constrained by observations, Nature, 528, 115-118, 2015.

---

## Referee Comment (RC2) · Anonymous Referee #2 · 15 Nov 2016

Summary and comments on the manuscript entitled

**A high-resolution synthetic bed elevation of the Antarctic continent**

presented on 04.07.2016

by

F. S. Graham et al.

**Summary & Review**

The authors present a statistical approach to create a synthetic map of the basal topography beneath the Antarctic ice sheet. The map is a superposition of a low-pass filtered Bedmap2 topography and a high-pass filtered synthetic roughness terrain (RT). The unfiltered RT is a random sample created such that the terrain roughness value and the covariance structure are preserved from the input thickness measurements. The used radar measurements are a compilation from Bedmap1 input data and the ICECAP campaigns. The approach is most valuable in areas where we have no information on bedrock topography, because statistical characteristics are preserved in the synthetic bed topography. In their abstract and the introduction, the authors state that the 'bed elevation is one of the most important controls in modelling ice-sheet dynamics'. Therefore, a 'detailed knowledge of the topography, [...], is required'. I could not agree more. This implies that a valuable bed topography map should reflect the measurements because there thicknesses are know. Yet, the big disadvantage of this map is that it fails to reproduce (within certain bounds) ice thicknesses where measurements were in fact available and used to infer the roughness and the covariance. This is important as the basal topography is a dominant source for modelling uncertainties, certainly on Antarctica. For me, this issue raises the question of what purpose the presented synthetic map can have to the modelling community because observed features are not present in this map. The authors argue that the map is useful for model sensitivity studies especially with respect to model resolution. In light of this limitation, the abstract seems to insinuate much more: '[...], the simulated bed elevation terrain has applicability in high-resolution ice-sheet modelling studies, including investigations of the interaction between topography, ice-sheet dynamics, and hydrology, where processes are highly sensitive to bed elevations.' My conclusion is therefore that I have to suggest that the manuscript should at least undergo a major revision.

The manuscript is well written and to the point.

**Main comments**

**Unconditioned simulation**

I appreciate that you mention that the map 'does not necessarily honour the exact values of the original data' and is 'not intended as a substitute for Bedmap2'. The fact that observations are not reproduced is deliberate, as you chose a non-conditional approach that will produce random thickness differences to the observations (dependent on the stochastic realisation). I wonder now if it is feasible for you to follow a conditional approach where you can constrain the bedrock elevation values to the observational input (optionally accounting for the uncertainties in the thickness measurements). Could this be solved by not using a uniform random matrix with zero mean to create the CDRT but an appropriate choice of this matrix? I fear that this might be a highly under-determined problem. An alternative could be that you drag back your synthetic terrain to the observations during the roughness scaling. I could think of some Gaussian function around the observations which force back the values to the observations. The downside is that neither the covariance structure nor the roughness will exactly be preserved from the original data. From a modelling perspective however, a conditioned map would be more valuable and could even be perceived as a regional alternative to Bedmap2. If this should not be feasible, you have to adjust the manuscript so that it becomes much clearer that the map does not necessarily reproduces the observations and that its advantages are in areas where no observations are available. In these areas, I wonder how this synthetic map would compare to other thickness reconstruction approaches (for instance the thickness reconstruction for the Peninsula from Huss et al. , 2014, doi:10.5194/tc-8-1261-2014).

**Base-map and coverage**

Another disadvantage is that the authors decided for Bedmap2 as the low-frequency base-map. This choice is certainly good in areas where no observations were available. In areas with observations, this implies that the synthetic map often shows a large mismatch to the observation used for inferring statistical measures. The reason why the authors relied on the Bedmap2 base-map is in my view twofold. First, they decided to present a high-resolution map for the whole of Antarctica. Second, the coverage of the available/used observations is sparse or even zero over large areas of the continent. As I said above, you have high discrepancies between the observations and Bedmap2, which are not accounted for in your approach. I wonder if it was not more consistent to adjust the base-map relying on your observations (your base-map shows anyhow a 10 times smaller grid spacing). This could be a simple interpolation between measurements and neighbouring Bedmap2 points. This guarantees that your final product shows a better match to the measurements. Your data coverage seems sufficient to do this in certain regions. If you decided for a more regions product, you have to clearly state differences of your approach to Goff et al., (2014). If you want to apply your approach to the entire continent, you might need more measurements. No matter which way you go, it

seems essential to me that your approach better reproduces the observations where they are available.

**Specific comments**

Figure 1  The abbreviation CDRT is not defined when you first refer to Figure 1 (P2L31). What are the numbers? In my printed version the drainage basin lines vanished. A shading could help.

Figure 2  Could you add Bedmap2 in all panels as a reference. That would help to assess the improvements or differences.

Figure 4  It is very difficult to assess this figure as the reference statistics of Bedmap2 are not given. It might be worth adding an accompanying figure with the same statistics for Bedmap2. The distributions might simply be dominated by Bedmap2 and difference would not be prominent. Then you might want to compare the distributions when you subtract the low-pass filtered Bedmap2 topography from both the original Bedmap2 and the synthetic HRES map.

---

## Author Comment (AC1) · 16 Mar 2017

Response to the reviewers on the ESSD manuscript (essd-2016-18) entitled

**A high-resolution synthetic bed elevation of the Antarctic continent**
Authors: Felicity S. Graham, Jason L. Roberts, Ben K. Galton-Fenzi, Duncan Young, Donald Blankenship, and Martin J. Siegert

**General comment**
We thank both reviewers for their comments that have substantially improved the clarity and accuracy of the manuscript.

Our response to both reviewers are presented below. In each case, the reviewer's comments are italicised. We have also attached a revised manuscript with changes highlighted.

**Reviewer 1 comments**
1.  *First, it is evident that "topographic variance" is not a smoothly varying function spatially. This is problematic because the high-frequency component of the topography has only been sampled for a small fraction of the continent (Fig 1). The high frequency characteristics of almost all of West Antarctica and most of East, remain uncharted. In addition, the fraction sampled does not incorporate many of the marine sectors (particularly in West Antarctica) that are highlighted in the introduction as motivation for the work.*

    The reviewer raises a very important point. We agree that topographic variance is not a smoothly varying function. We also agree that the ability to generate a high-resolution terrain that reproduces the observations with the approach taken in this paper is contingent on having high resolution data from which to resolve the high frequency component. We highlight this limitation in our approach, including the assumption we have made with respect to topographic variance, in the updated manuscript, page 7, lines 1-3, as follows:

    > "In order to generate the high-frequency roughness terrain, we assume that topographic variance is a smoothly varying function spatially. In reality, we have too few high-resolution data points to adequately assess the rigour of such an assumption."

    One of the motivations for the generation of HRES is the lack of high resolution data for the Antarctic continent. We acknowledge the imperfections of HRES in terms of its fidelity to observations; however, HRES does offer considerably more in terms of understanding interactions between ice dynamics and bedrock roughness than other datasets than are currently available. Logistically, it will take a long time to collect the data needed to generate a high-resolution dataset that preserves the observations on the scale of HRES.

However, the aim of this work was not to produce a high-resolution dataset that exactly matches observations. Rather, the aim was to generate a high-resolution dataset that has some fidelity to observations, and with which we can begin to address the question of the importance of resolution on ice dynamics. In order to have a self-consistent dataset, we were only able to include high-resolution observations that satisfied the criteria in section 2.1.1 (page 4), which excluded many low-resolution samples, including in West Antarctica.

The priority areas in generating HRES, and those in which the most data were available to the authors, were those marine-based regions surveyed by ICECAP in East Antarctica. We recognise that the fidelity of HRES to the observations here will be sub-optimal due to our assumption of smoothly varying topography. Nevertheless, this is the best we can do using the current approach.

We are currently collecting other high resolution data products in regions that were not well charted in the current version of HRES and will include them in subsequent versions. We note (page 3, lines 2-3) that the approach taken in this paper allows us to easily improve the high-resolution data coverage at the regional scale as more data are made available.

2. *Second, the authors refer to "topographic roughness" without being clear about precisely what length scale this relates to. Inherently this length scale is determined by the along track sampling properties of the radar system rather than any underlying geophysical criteria. This is problematic because a key uncertainty in future projections is basal traction (Ritz et al, 2015), which is modulated by metre scale roughness. It would appear that HRES provides no information on roughness at this scale or at any scale below 200 m (given 100 m bin size). While this is shorter wavelength than BEDMAP2 it remains to demonstrated that it is adequate to elucidate the role and/or importance of "bed roughness" on ice dynamics.*

We agree with the reviewer that HRES has no information on metre scale roughness -- or indeed on any scale below 200m – which is potentially important in governing Antarctic flow dynamics. Given that our criterion for including high resolution data in the simulation of HRES was that the data were at least 100m resolution, the covariance matrix generated for the simulation of the CDRT matrix will not have information below the 200m length scale. This certainly limits our capacity to investigate the role of fine scale (i.e., 100m and finer) bed roughness on ice dynamics. We have modified the manuscript to make the length scale used in this study much more explicit. See updates to the manuscript in the introduction, page 2 lines 25-26:

"The length scale of the topographic roughness used in this study is limited to 200m."

and section 4, page 7 point (vii):

"It is possible that even finer scale topographic features than those captured in HRES play a role in modulating ice dynamic processes (e.g., Ritz et al., 2015). This has

implications for the degree to which future modelling will ascertain what resolution in bed topography is enough for consistent and accurate simulations of ice dynamics (i.e., we can only assess the impact of bed topography features of a scale greater than 100m). We will explore this further in subsequent studies."

A subsequent high-resolution modelling study will investigate whether 100m resolution is sufficient, and may provide an indication of the length scales that are the dominant controls on ice dynamics. We also note that for a whole of Antarctica dataset, storage, availability, and usability for many people becomes an issue (for example, a 10m resolution dataset would be 1.7TB).

3. *These issues are challenging to address, requiring greater data coverage, which does exist, and sensitivity studies with an ice sheet model to scales of basal topographic variability.*

The current intent is to use HRES in sensitivity studies that address the impact of resolution (and hence basal topographic variability) on ice sheet dynamics in regions of East Antarctica, discussed in the updated manuscript, section 6, page 8, lines 9-12:

"The sufficiency of the resolution of HRES for addressing the sensitivity of ice-sheet dynamic processes to bed elevation resolution will be addressed in a subsequent numerical modelling study. The results of the modelling study will also emphasise regions where high-resolution bed elevation data are needed, which will facilitate targeted efforts in data collection."

The issue of improving the data coverage – particularly in regions of West Antarctic – is a focus of current attention. HRES will be updated as more data are made available, as per page 7 point (iii):

"Only ICECAP/BC1 data of sufficiently high-resolution (i.e., greater than 100m resolution, chosen as it is twice the Nyquist frequency of the observations) were included in the simulation of HRES. This limits how well the final HRES dataset matches the observations, especially in regions of West Antarctica. The roughness terrain will be updated to incorporate additional high-resolution bed elevation data as they become available."

**Specific comments**
4. *p2, l7 "heavy smoothing" is non scientific terminology. What is heavy? It doesn't have a mass.*
"Heavy smoothing" has been removed.

5. *p2, l8 poor phrasing.*
Original text:
"The purpose of this study is to generate a high-resolution synthetic bed topography dataset for Antarctica (HRES) with small-scale roughness incorporated, matched to that measured in available radar transects."

Modified (page 2, lines 18-22), as follows:
"The purpose of this study is to generate a high-resolution synthetic bed topography dataset for Antarctica (HRES) for investigating the sensitivity of ice-sheet dynamics to bed elevation resolution, including the interaction with subglacial hydrology (Fricker et al., 2007; Goff et al., 2014)."

6. *p2 l16-17 ditto*
Original text:
"Importantly, the question remains as to the minimum degree of spatial resolution required in bed topography DEMs to accurately model ice flow."

Modified (page 2, lines 15-17), as follows:
"Importantly, a question that has yet to be addressed for the Antarctic continent is what minimum resolution in bed elevation is required to accurately simulate ice-sheet dynamics."

7. *p4, l15 artefacting not a word*
Removed "artefacting" throughout the manuscript

8. *Fig 3. Figs a and b essentially identical (to the eye) and provide no real insight: they are identical to the BEDMAP2 topography at the scale plotted. Fig c appears to have had something horrible go wrong with the colour table/conversion. I have no idea what value the purple colours are. This figure needs redrafting to provide some useful info.*

The authors agree that the small size of figure 3 panels in the original manuscript make the comparison between Bedmap2 and HRES difficult. However, there are clear differences between the bed elevations illustrated in panels a (for Bedmap2) and b (for HRES). We have modified figure 3 by enlarging the panel size so that the differences between Bedmap2 and HRES are more apparent. The difference plot (panel c) has been modified to show the absolute difference between Bedmap2 and HRES with a new colour scale.

9. *Fig 5. Even when zoomed to larger than the printed page I struggled to read the numbers and see the detail in the 18 graphs in this figure.*

Figure 5 text and lines have been enlarged to make the details clearer.

**Reviewer 2 comments**
1. *In their abstract and the introduction, the authors state that the 'bed elevation is one of the most important controls in modelling ice-sheet dynamics'. Therefore, a 'detailed knowledge of the topography, [...], is required'. I could not agree more. This implies that a valuable bed topography map should reflect the measurements because there thicknesses are know. Yet, the big disadvantage of this map is that it fails to reproduce (within certain bounds) ice thicknesses where measurements were in fact available and used to infer the roughness and the covariance. This is important as the basal*

*topography is a dominant source for modelling uncertainties, certainly on Antarctica. For me, this issue raises the question of what purpose the presented synthetic map can have to the modelling community because observed features are not present in this map. The authors argue that the map is useful for model sensitivity studies especially with respect to model resolution. In light of this limitation, the abstract seems to insinuate much more: '[...], the simulated bed elevation terrain has applicability in high-resolution ice-sheet modelling studies, including investigations of the interaction between topography, ice-sheet dynamics, and hydrology, where processes are highly sensitive to bed elevations.'*

In the introduction, we write that "a detailed knowledge of the topography…is required". However, an outstanding question in the glaciology community is to what degree do we need to know the resolution of the bedrock topography in order to get consistent and accurate models of ice sheet dynamics (page 2, lines 15-17). To answer such a question, we first need a high-resolution bed topography dataset, which is not currently available given the paucity of data across most of the Antarctic continent.

Hence, the overall intent of this work was to produce a dataset that can be used to address the question of the sensitivity of ice dynamic processes to underlying bed topography resolution. For the purposes of a modelling sensitivity study, and consistent with previous studies (e.g., Durand et al., 2011), it is not necessary to produce a high-resolution dataset that matches the observations exactly, but rather to produce a synthetic dataset that has similar characteristics (i.e., covariance) to the observations. The results of a subsequent modelling sensitivity study will indicate regions where we should focus future field campaigns for data collection.

In the updated manuscript, we have emphasised that HRES was generated for this purpose in the introduction, page 2, lines 18-22:

> "The purpose of this study is to generate a high-resolution synthetic bed topography dataset for Antarctica (HRES) for investigating the sensitivity of ice-sheet dynamics to bed elevation resolution, including the interaction with subglacial hydrology 20 (Fricker et al., 2007; Goff et al., 2014). We emphasise that this dataset is intended to be synthetic (i.e., HRES is not intended to be a substitute for other bed elevation datasets that preserve the observations), but has covariance properties that are consistent with those of the measured bed elevations from available radar transects."

and section 6 (page 8, lines 6-12):

> "HRES is not intended as a realistic depiction of high-resolution Antarctic bed topography and is, therefore, not meant as a substitute for datasets such as Bedmap2 (although, at resolutions > 5 km, HRES is identical to the Bedmap2 bed elevation dataset). Instead, HRES is a synthetic terrain generated for the specific purpose of assessing the sensitivity of ice-sheet dynamic processes to the resolution of the underlying bed topography. The sufficiency of the resolution of HRES for addressing the sensitivity of ice-sheet dynamic processes to bed elevation resolution

will be addressed in a subsequent numerical modelling study. The results of the modelling study will also emphasise regions where high-resolution bed elevation data are needed, which will facilitate targeted efforts in data collection."

reiterating that our dataset is not intended as a substitute to "realistic" bed topography datasets, such as Bedmap2.

**Main comments**
2. *Unconditioned simulation*
   *I appreciate that you mention that the map 'does not necessarily honour the ex- act values of the original data' and is 'not intended as a substitute for Bedmap2'. The fact that observations are not reproduced is deliberate, as you chose a non- conditional approach that will produce random thickness differences to the observations (dependent on the stochastic realisation). I wonder now if it is feasible for you to follow a conditional approach where you can constrain the bedrock elevation values to the observational input (optionally accounting for the uncertainties in the thickness measurements).*

   We deliberately did not choose a method of conditional simulation (e.g., Goff et al., 2010) because it would have relied on having regularly spaced data that are of higher resolution than the data available to us. At a regional scale, such a type of simulation is feasible, but not for the whole Antarctic continent. The irregularity and spatial inhomogeneity of the observational data preclude a conditional simulation, which was also not the focus of the study.

   *Could this be solved by not using a uniform random matrix with zero mean to create the CDRT but an appropriate choice of this matrix? I fear that this might be a highly under-determined problem.*

   It might be possible to use a different choice of matrix (e.g., a different distribution with different mean value), but again, this would require significantly more data at the scale we are interested in than are currently available.

   *An alternative could be that you drag back your synthetic terrain to the observations during the roughness scaling. I could think of some Gaussian function around the observations which force back the values to the observations. The downside is that neither the covariance structure nor the roughness will exactly be preserved from the original data.*

   With the method of Cholesky decomposition, we cannot alter the data to match the observations and at the same time ensure that the covariance properties are preserved. While such an approach would enforce conditionality, it would come at the expense of biasing the roughness, which would contradict the aims of the study to generate a dataset to investigate the impact of topographic roughness on ice sheet dynamics.

   *From a modelling perspective however, a conditioned map would be more valuable and could even be perceived as a regional alternative to Bedmap2.*

In regions where observational data coverage is better, a conditional simulation like the one discussed in Goff et al. (2010) might be an improvement. For regions where data are not dense, or regularly spaced, we must rely on other methods. The unconditional simulation used here is the best option available to us given our aim to generate a high-resolution, synthetic, whole of Antarctica dataset to investigate the sensitivity of ice sheet dynamic processes to bedrock roughness resolution. As we make clear in the updated manuscript, our intention is not to replace Bedmap2, but to provide a suitable dataset to investigate the effect of roughness on ice sheet dynamics, as per the updated manuscript, page 2, lines 18-22:

"The purpose of this study is to generate a high-resolution synthetic bed topography dataset for Antarctica (HRES) for investigating the sensitivity of ice-sheet dynamics to bed elevation resolution, including the interaction with subglacial hydrology 20 (Fricker et al., 2007; Goff et al., 2014). We emphasise that this dataset is intended to be synthetic (i.e., HRES is not intended to be a substitute for other bed elevation datasets that preserve the observations), but has covariance properties that are consistent with those of the measured bed elevations from available radar transects."

*If this should not be feasible, you have to adjust the manuscript so that it becomes much clearer that the map does not necessarily reproduces the observations and that its advantages are in areas where no observations are available. In these areas, I wonder how this synthetic map would compare to other thickness reconstruction approaches (for instance the thickness reconstruction for the Peninsula from Huss et al. , 2014, doi:10.5194/tc-8-1261-2014).*

It is indeed the aim of this project to generate a bed topography map that does not reproduce the observations exactly. We have modified the abstract (page 1), section 1 (page 2, lines 18-22), and section 6 (page 8, lines 6-12) to make this point clearer.

3. *Base-map and coverage*
   *Another disadvantage is that the authors decided for Bedmap2 as the low-frequency base-map. This choice is certainly good in areas where no observations were avail- able. In areas with observations, this implies that the synthetic map often shows a large mismatch to the observation used for inferring statistical measures.*

The authors are aware of the shortcomings of the methods used to generate Bedmap2, particularly those methods that introduce aliasing and inaccuracies in regions where observations are not available, as discussed in the updated manuscript, section 4, page 7, point (iv):

> "The Bedmap2 DEM, of which the low-pass component is included in the generation of HRES, suffers from artefacts through the particular gridding and interpolating methods used compared with other ice thickness interpolation methods, especially in regions with no nearby measurements (Roberts et al., 2011)."

Nevertheless, Bedmap2 has been widely used in the glaciological community for modelling studies. Given the intent of our work to investigate the sensitivity of ice

dynamic processes to the resolution of the underlying bed topography, and that HRES therefore does not necessarily reproduce observed bed topographies, Bedmap2 serves as a sufficient choice of low-frequency bed topography.

*If you want to apply your approach to the entire continent, you might need more measurements.*

The acquisition of further measurements to improve HRES is ongoing and HRES will be updated as such data are made available to the authors. This is discussed in section 2 (pages 3, lines 2-3) and section 4 (page 7, point iii):

"Only ICECAP/BC1 data of sufficiently high-resolution (i.e., greater than 100 m resolution, chosen as it is twice the Nyquist frequency of the observations) were included in the simulation of HRES. This limits how well the final HRES dataset matches the observations, especially in regions of West Antarctica. The roughness terrain will be updated to incorporate additional high-resolution bed elevation data as they become available."

*It seems essential to me that your approach better reproduces the observations where they are available.*

As discussed above, and given the paucity of data for the whole Antarctic continent, a non-conditional high-resolution bed topography dataset is sufficient for our purposes to investigate the sensitivity of ice dynamic processes to the underlying bed topography resolution.

**Specific comments**

4. *In figure 1, the abbreviation CDRT is not defined when you first refer to Figure1 (P2L31). What are the numbers? In my printed version the drainage basin lines vanished. A shading could help.*

CDRT has now been defined and the caption to figure 1 updated. The numbers are the drainage basins derived from the Goddard Ice Altimetry Group from ICESat data (as per the updated caption) and correspond with the drainage basins referred to in figures 3-5. Shading lines have been updated.

5. *Figure 2. Could you add Bedmap2 in all panels as a reference. That would help to assess the improvements or differences.*

This figure demonstrates that once CDRT is suitably scaled, the distributions of HRES and the underlying observations used to generate the high frequency component of HRES are similar. Given the limited spatial resolution of Bedmap2, such a comparison with the Bedmap2 would offer no insight.

6. *Figure 4. It is very difficult to assess this figure as the reference statistics of Bedmap2 are not given. It might be worth adding an accompanying figure with the same statistics for Bedmap2. The distributions might simply be dominated by Bedmap2 and difference*

*would not be prominent. Then you might want to compare the distributions when you subtract the low-pass filtered Bedmap2 topography from both the original Bedmap2 and the synthetic HRES map.*

Although correctly referred to in the text of the manuscript, the original caption on this figure incorrectly labelled the blue distributions as those belonging to HRES. In fact, this figure shows how different from the normal distribution is the difference between the Bedmap2 and HRES (i.e., $D$=Bedmap2-HRES) bed elevations for each of the drainage basins. As discussed on page 5 lines 20-21, the degree to which $D$ differs from the normal distribution provides a measure of the fidelity of HRES to the original ICECAP/Bedmap1 data. As such, including either distributions of Bedmap2 or HRES bed elevations for each of the drainage basins would offer no additional insight. The caption has been corrected in the updated manuscript.

**A high-resolution synthetic bed elevation grid of the Antarctic continent**

Felicity S. Graham[1], Jason L. Roberts[2,3], Ben K. Galton-Fenzi[2,3], Duncan Young[4], Donald Blankenship[4], and Martin J. Siegert[5]

[1]Institute for Marine and Antarctic Studies, University of Tasmania, Private Bag 129, Hobart, Tasmania 7001, Australia
[2]Australian Antarctic Division, Kingston, Tasmania, Australia
[3]Antarctic Climate and Ecosystems Cooperative Research Centre, Private Bag 80, Hobart, Tasmania 7001, Australia
[4]Institute for Geophysics, University of Texas at Austin, Austin, Texas 78758, USA
[5]Grantham Institute and Department of Earth Sciences and Engineering, Imperial College London, London SW7 2AZ, UK

*Correspondence to:* F. S. Graham (felicity.graham@utas.edu.au)

**Abstract.** Digital elevation models of Antarctic bed topography are  smoothed and interpolated onto low-resolution ($> 1$ km) grids as  current observed topography data are generally sparsely and unevenly sampled. This issue has potential implications for numerical simulations of ice-sheet dynamics, especially in regions prone to instability where detailed knowledge of the topography  including fine-scale roughness  is required. Here, we present a high-resolution (100 m) synthetic bed elevation terrain for  Antarctica, encompassing the continent, continental shelf, and seas south of 60°S. The synthetic bed surface – denoted HRES – preserves topographic roughness characteristics of airborne and ground-based ice-penetrating radar data  measured by the ICECAP consortium or used to create the Bedmap1  compilation. At broader scales ($> 5$ km resolution), HRES is identical to the Bedmap2 bed elevation data. HRES has applicability in high-resolution ice-sheet modelling studies, including investigations of the interaction between topography, ice-sheet dynamics, and hydrology, where processes are highly sensitive to bed elevations and fine-scale roughness. The data are available for download  from the Australian Antarctic Data Centre (doi:10.4225/15/57464ADE22F50).

**1 Introduction**

Estimates of mass loss from both the Antarctic and Greenland Ice Sheets are associated with the largest degree of uncertainty in projections of sea level rise over the coming century (Church et al., 2013). As the most vulnerable regions of the Antarctic Ice Sheet are grounded below sea level, the ice-sheet response to climate warming will be determined by dynamics operating at the grounding line (Schoof, 2007b; Drouet et al., 2013).  Where the bed topography slopes downward into the interior of  marine-based ice-sheets, Marine Ice Sheet Instability (MISI) could occur, leading to increased ice flow, thinning, and rapid glacier retreat (Weertman, 1974; Thomas et al., 2004; Schoof,

2007a; Durand et al., 2009; Goldberg et al., 2009; Favier et al., 2014; Joughin et al., 2014). It follows that bed elevation is one of the most important controls in modelling ice-sheet dynamics and constraining estimates of future sea level rise.

Concerted international efforts over recent decades have vastly increased the scope and density of bed elevation measurements in Antarctica (Lythe et al., 2001; Le Brocq et al., 2010; Fretwell et al., 2013). These data have been used to improve the fidelity of gridded digital elevation models (DEMs) spanning the whole Antarctic continent. Building on a 5 km gridded bed elevation DEM (Lythe et al., 2001; Le Brocq et al., 2010), the most recently compiled Antarctic bed topography dataset, Bedmap2, is available at 1 km resolution, having been generated from over 25 million measurements (Fretwell et al., 2013). Nevertheless, much of the Antarctic continent is difficult to access  and remains poorly sampled. In such regions, bed elevation DEMs rely on interpolation, resulting in geometric inconsistencies that  adversely impact numerical simulations of ice dynamics (Warner and Budd, 2000; Fürst et al., 2015; Gasson et al., 2015). Uncertainties in bed elevation are particularly problematic given that, for much of the Antarctic Ice Sheet, the simulated large-scale velocity field depends only on the  local-scale details of the geometry and boundary conditions due to the elliptic nature of the governing equations for ice flow.

Recent effort has focussed on understanding the impact of low-resolution bed elevation data on ice mass-flux. Durand et al. (2011) performed a sensitivity analysis of an outlet glacier susceptible to MISI, demonstrating that  at least 1 km spatial resolution in bed topography is required for accurate estimates of ice mass flux. However, bed elevation data of a higher resolution than 1 km may be necessary in some applications to capture both the channelised landscape that guides glacier flow and the fine-scale roughness that impacts basal sliding  (Goff et al., 2014; Ritz et al., 2015) . Importantly, a question that has yet to be addressed for the Antarctic continent is what minimum resolution in bed elevation is required to accurately simulate ice-sheet dynamics.

The purpose of this study is to generate a high-resolution synthetic bed topography dataset for Antarctica (HRES)  for investigating the sensitivity of ice-sheet dynamics to bed elevation resolution, including the interaction with subglacial hydrology (Fricker et al., 2007; Goff et al., 2014) . We emphasise that this dataset is intended to be synthetic (i.e., HRES is not intended to be a substitute for other bed elevation datasets that preserve the observations), but has covariance properties that are consistent with those of the measured bed elevations from available radar transects. The generation of HRES relies on bed elevation data used to create the Bedmap1 compilation and from the ICECAP airborne radar survey  where they are available at high-resolution.  The low-resolution ($>$ 5 km) component of HRES is identical to Bedmap2. HRES covers the same domain as Bedmap2 and is available at a spatial resolution of 100 m. The length scale of the topographic roughness used in this study is limited to 200 m.

**2 Data synthesis**

A two-step approach was used to generate the high-resolution synthetic bed elevation terrain, HRES. , as follows. First, we simulated a non-conditional "roughness" terrain (i.e., a stochastic realisation of "roughness" that does not necessarily honour the exact values of the original data) using high spatial resolution radar data obtained from the 2009-2012 ICECAP cam-

5  paigns  (Roberts et al., 2011; Young et al., 2011; Wright et al., 2012; Bla... used to create the Bedmap1 compilation  (hereafter BC1; Lythe et al., 2001) . The locations of the data included in this step are  shown in Fig. 1. The ICECAP bed elevation data are measured using a High-Capability Radar Sounder (HiCARS) high bandwidth airborne ice penetrating radar (Peters et al., 2005);  BC1 combines data from multiple airborne and ground based radar sounding campaigns, from a variety of systems. Our method for

10  the generation of the roughness terrain can easily incorporate additional bed elevation data  as they become available. Once generated, the roughness terrain was high-pass filtered using a gaussian kernel with a 5 km 1/2 power cutoff.

Second, the Bedmap2 bed topography DEM was low-pass filtered, using a low-pass gaussian kernel with a 5 km 1/2 power cutoff. The two filtered terrains were combined (preserving all wavelengths of the original datasets), resulting in the high-resolution bed topography, HRES.

15  An alternative method for the production of high (250 m) resolution bed elevation data has recently been applied to the Thwaites Glacier region (Goff et al., 2014) . Goff et al. (2014) combines both conditional and non-conditional simulations of a range of data with the intent to avoid the inconsistencies and artefacts introduced through interpolation techniques such as kriging. The resulting terrain is of sufficiently high-resolution to facilitate characterisation of the subglacial landforms and landscape of the Thwaites Glacier, which will lead to improved understanding of ice flow and its sensitivities to external forcing.

20  However, the methods used to produce this terrain rely on a higher data density than is available for most of Antarctica. Our methodology was chosen because of its ability to handle spatially sparse data with highly inhomogeneous sampling resolutions, while being computationally tractable for all of Antarctica at 100 m resolution.

In the following sections, we provide a detailed outline of the methods used to generate the roughness terrain and to compile the final synthetic bed topography dataset. The 'pseudo' algorithm for the generation of HRES is provided in Appendix A.

**2.1 Roughness terrain synthesis**

Ideally, the spatial covariance characteristics of the non-conditional roughness terrain (the high frequency component of the synthetic topography dataset) should match those of ICECAP and BC1. The method of Cholesky decomposition of the observed covariance matrix can be used to produce such correlated data (Davis, 1987). Specifically, the positive definite covariance matrix $\mathbf{C}$ calculated from the ICECAP and  BC1 datasets can be decomposed

30  into lower $\mathbf{L}$ and upper $\mathbf{U}$ triangular matrices

$$\mathbf{C} = \mathbf{LU}, \tag{1}$$

where $\mathbf{L}$ has real and positive diagonal entries and $\mathbf{U}$ is the conjugate transpose of $\mathbf{L}$. This method results in a unique decomposition for positive definite matrix $\mathbf{C}$.

Now, given a vector $\mathbf{z}$ of uniformly distributed random numbers with zero mean, we find

$$\text{Cov}(\mathbf{Lz}) = E[(\mathbf{Lz})(\mathbf{Lz})^T] = E(\mathbf{Lzz}^T\mathbf{U}) = \mathbf{LIU} = \mathbf{C}. \tag{2}$$

[revised manuscript text omitted]

(iii) Only ICECAP/BC1 data of sufficiently high-resolution (i.e., greater than 100 m resolution, chosen as it is twice the Nyquist frequency of the observations) were included in the simulation of HRES. This limits how well the final HRES dataset matches the observations, especially in regions of West Antarctica. The roughness terrain will be updated to incorporate additional high-resolution bed elevation data as they become available.

[revised manuscript text omitted]

*Acknowledgements.* The authors thanks Richard Coleman and David E. Gwyther for constructive feedback and two anonymous reviewers for their suggestions that improved the manuscript. This research is supported under the Australian Research Council's Special Research Initiative for Antarctic Gateway Partnership SR140300001. The project is part of an ongoing ICECAP collaboration between Australia, the USA, and the UK, and is supported by the Australian Antarctic Division projects 3013, 4077 and 4346, the Antarctic Climate and Ecosystems Cooperative Research Centre, NSF grants PLR-0733025 and PLR-1143843, and CDI-0941678, NASA grants NNG10HPO6C and NNX11AD33G (Operation Ice Bridge and the American Recovery and Reinvestment Act), NERC grant NE/D003733/1, the G. Unger Vetlesen Foundation, the Jackson School of Geosciences, University of Texas, and the British Council Global Innovation Initiative Award. This is UTIG contribution ####.

[revised manuscript text omitted]

C

**Figure 3.**  (c) Absolute difference between Bedmap2 and HRES  bed elevations (m). The drainage  basins (black lines) are taken from the Goddard Ice Altimetry Group from ICESat data see Fig. **??**http://icesat4.gsfc.nasa.gov/ cryo_data/ant_grn_drainage_systems.php

[Figure]

**Figure 4.** Distribution of the difference between Bedmap2 and HRES ($D =$ Bedmap2 − HRES) over the Antarctic drainage divides from the Goddard Ice Altimetry Group from ICESat data. Basins 2-17 are in East Antarctica, basins 1 and 18-23 are in West Antarctica, and the remaining basins are located in the Antarctic Peninsula (Fig. 1). The blue binned data are $D$ and the red dashed lines show the normal distribution from the given mean and standard deviation of $D$.

[Figure]

**Figure 5.**  (a) Locations of selected flight lines from the ICECAP/ BC1 compilations.  (b) Along track bed elevations from ICECAP/ BC1 (black) and corresponding overlay points from Bedmap2 (blue) and HRES (green). The x-axis shows the along track distance (km) from the first point in the flight line.

**Table 1.** Statistics from the D'Agnostino-Pearson Agostino-Pearson $K^2$ normality test Eq. (5) for each of the drainage basins 1-27 in Fig. 4. The $\sqrt{b_1}$ and $b_2$ statistics are the bases for tests of skewness and kurtosis, respectively. For a normal distribution, $K^2$ is approximately chi-distributed with two degrees of freedom.

| Basin | $\sqrt{b_1}$ | $b_2$ | $Z(\sqrt{b_1})$ | $Z(b_2)$ | $K^2$ | $p$ |
|---|---|---|---|---|---|---|
| 1 | -0.01 | 3.09 | -2.16 | 14.82 | 224.33 | 0.00 |
| 2 | 0.01 | 4.84 | 2.95 | 193.11 | 37300.76 | 0.00 |
| 3 | 0.00 | 6.34 | 0.18 | 347.69 | 120888.51 | 0.00 |
| 4 | 0.00 | 4.81 | 0.57 | 115.23 | 13279.22 | 0.00 |
| 5 | -0.02 | 4.37 | -4.95 | 83.48 | 6992.95 | 0.00 |
| 6 | -0.02 | 3.58 | -7.44 | 77.31 | 6031.82 | 0.00 |
| 7 | -0.02 | 3.93 | -5.45 | 93.50 | 8771.66 | 0.00 |
| 8 | 0.06 | 3.85 | 10.61 | 49.83 | 2595.36 | 0.00 |
| 9 | 0.03 | 4.03 | 5.20 | 57.94 | 3384.57 | 0.00 |
| 10 | 0.03 | 4.70 | 12.62 | 187.51 | 35319.39 | 0.00 |
| 11 | 0.01 | 3.91 | 1.47 | 67.42 | 4547.69 | 0.00 |
| 12 | 0.01 | 6.68 | 2.74 | 257.46 | 66290.96 | 0.00 |
| 13 | 0.02 | 8.56 | 8.39 | 365.49 | 133654.10 | 0.00 |
| 14 | 0.02 | 5.51 | 7.82 | 207.25 | 43014.01 | 0.00 |
| 15 | 0.12 | 3.87 | 18.39 | 46.05 | 2458.58 | 0.00 |
| 16 | 0.02 | 7.31 | 4.26 | 160.71 | 25845.09 | 0.00 |
| 17 | 0.09 | 6.20 | 52.45 | 387.82 | 153152.39 | 0.00 |
| 18 | 0.02 | 5.14 | 4.87 | 140.09 | 19650.24 | 0.00 |
| 19 | -0.01 | 3.62 | -3.41 | 66.41 | 4422.35 | 0.00 |
| 20 | 0.00 | 3.20 | 0.58 | 18.21 | 331.76 | 0.00 |
| 21 | 0.01 | 3.02 | 1.13 | 2.18 | 6.03 | 0.05 |
| 22 | -0.00 | 3.14 | -0.63 | 12.13 | 147.46 | 0.00 |
| 23 | 0.01 | 3.05 | 1.71 | 3.45 | 14.82 | 0.00 |
| 24 | -0.06 | 7.04 | -10.77 | 139.75 | 19646.67 | 0.00 |
| 25 | 0.00 | 3.34 | 0.32 | 11.44 | 130.89 | 0.00 |
| 26 | -0.01 | 4.88 | -1.75 | 67.36 | 4541.02 | 0.00 |
| 27 | -0.02 | 4.17 | -2.00 | 40.79 | 1667.76 | 0.00 |

**Table 2.** Along track roughness (m) from the ICECAP/ BC1 flight lines in Fig. 5 and the corresponding roughness values from the HRES and Bedmap2 overlay points. Root mean square errors (rmse; m) between the ICECAP/ BC1 data and the corresponding HRES and Bedmap2 data were normalised by the square root of the number of points in each track. For the ICECAP flight lines, the unique PST (Project/Season/Track) identifier is reported; for the BC1 flight lines, the mission number is reported.

|  | source | identifier | ICECAP/ BC1 roughness (m) | HRES roughness (m) | rmse (m) | Bedmap2 roughness (m) | rmse (m) |
|---|---|---|---|---|---|---|---|
| A | ICECAP | ASB/JKB1a/R13Ta | 134.8 | 82.7 | 6.6 | 41.5 | 3.8 |
| B | ICECAP | ASB/JKB1a/R21Wa | 166.2 | 177.8 | 2.4 | 64.9 | 1.5 |
| C | ICECAP | ASB/JKB1a/R13Wa | 83.6 | 77.8 | 1.0 | 30.9 | 0.8 |
| D | ICECAP | WSB/JKB1a/GL0263a | 107.3 | 85.9 | 3.2 | 25.3 | 3.1 |
| E | ICECAP | TRL/JKB2d/EX1EX2a | 121.9 | 113.5 | 3.4 | 36.2 | 2.7 |
| F | ICECAP | WSB/JKB2c/GL0233b | 134.5 | 93.1 | 5.5 | 74.6 | 4.0 |
| G | ICECAP | TRL/JKB2d/ES2TROa | 250.9 | 267.9 | 5.5 | 150.2 | 3.2 |
| H | ICECAP | WSB/JKB1a/GL0024b | 143.7 | 162.2 | 41.0 | 154.6 | 42.4 |
| I | ICECAP | WSB/JKB1a/GL0143a | 159.3 | 132.1 | 2.5 | 79.6 | 2.1 |
| J | ICECAP | ASB/JKB1a/Y07c | 118.9 | 117.2 | 5.2 | 25.4 | 5.2 |
| K | ICECAP | WSB/JKB1a/GL0373a | 72.7 | 75.4 | 1.8 | 32.1 | 1.4 |
| L | ICECAP | ASB/JKB2e/Y08b | 111.1 | 104.4 | 2.5 | 21.1 | 2.2 |
| M | ICECAP | WSB/JKB2e/GL0292c | 208.8 | 125.7 | 8.6 | 39.0 | 8.3 |
| N | ICECAP | ASB/JKB2h/R22Wa | 158.7 | 125.6 | 3.5 | 274.0 | 3.4 |
| O | ICECAP | ICP5/JKB2h/F09T01a | 75.7 | 30.2 | 5.9 | 21.5 | 6.1 |
| P | BC1 | 40 | 38.6 | 169.8 | 12.7 | 4.8 | 11.7 |
| Q | BC1 | 16 | 104.6 | 150.9 | 20.2 | 10.3 | 17.2 |
| R | BC1 | 21 | 10.0 | 270.0 | 11.8 | 34.7 | 9.5 |